# Pre-Emptive Use of Rituximab in Epstein–Barr Virus Reactivation: Incidence, Predictive Factors, Monitoring, and Outcomes

**DOI:** 10.3390/ijms242216029

**Published:** 2023-11-07

**Authors:** Apostolia Papalexandri, Eleni Gavriilaki, Anna Vardi, Nikolaos Kotsiou, Christos Demosthenous, Natassa Constantinou, Tasoula Touloumenidou, Panagiota Zerva, Fotini Kika, Michalis Iskas, Ioannis Batsis, Despina Mallouri, Evangelia Yannaki, Achilles Anagnostopoulos, Ioanna Sakellari

**Affiliations:** 1Hematology Department, BMT Unit, General Hospital “George Papanicolaou”, 57010 Thessaloniki, Greece; lila.papalexandri@gmail.com (A.P.); anna_vardi@yahoo.com (A.V.); christosde@msn.com (C.D.); tasoula.touloumenidou@gmail.com (T.T.); panagiotazerva@gmail.com (P.Z.); fotkika@gmail.com (F.K.); mic_iskas@yahoo.fr (M.I.); iobats@yahoo.gr (I.B.); dmallouri@gmail.com (D.M.); eyannaki@uw.edu (E.Y.); achanagh@gmail.com (A.A.); ioannamarilena@gmail.com (I.S.); 22nd Propedeutic Department of Internal Medicine, Aristotle University of Thessaloniki, 54642 Thessaloniki, Greece; kotsiounikolaos@gmail.com

**Keywords:** EBV reactivation, post-transplant lymphoproliferative disease, viral infection, hematopoietic stem cell transplantation, retrospective studies

## Abstract

Post-transplant lymphoproliferative disease (PTLD) is a fatal complication of hematopoietic cell transplantation (HCT) associated with the Epstein–Barr virus (EBV). Multiple factors such as transplant type, graft-versus-host disease (GVHD), human leukocyte antigens (HLA) mismatch, patient age, and T-lymphocyte-depleting treatments increase the risk of PTLD. EBV reactivation in hematopoietic cell transplant recipients is monitored through periodic quantitative polymerase chain reaction (Q-PCR) tests. However, substantial uncertainty persists regarding the clinically significant EBV levels for these patients. Guidelines recommend initiating EBV monitoring no later than four weeks post-HCT and conducting it weekly. Pre-emptive therapies, such as the reduction of immunosuppressive therapy and the administration of rituximab to treat EBV viral loads are also suggested. In this study, we investigated the occurrence of EBV-PTLD in 546 HCT recipients, focusing on the clinical manifestations and risk factors associated with the disease. We managed to identify 67,150 viral genomic copies/mL as the cutoff point for predicting PTLD, with 80% sensitivity and specificity. Among our cohort, only 1% of the patients presented PTLD. Anti-thymocyte globulin (ATG) and GVHD were independently associated with lower survival rates and higher treatment-related mortality. According to our findings, prophylactic measures including regular monitoring, pre-emptive therapy, and supportive treatment against infections can be effective in preventing EBV-related complications. This study also recommends conducting EBV monitoring at regular intervals, initiating pre-emptive therapy when viral load increases, and identifying factors that increase the risk of PTLD. Our study stresses the importance of frequent and careful follow-ups of post-transplant complications and early intervention in order to improve survival rates and reduce mortality.

## 1. Introduction

Post-transplant lymphoproliferative disease (PTLD) is one of the most fatal complications of hematopoietic cell transplantation (HCT) associated with the Epstein–Barr virus (EBV) in immunocompromised individuals [1,2]. EBV is a ubiquitous herpesvirus that infects over 90% of adults and 50–89% of children worldwide [1,3,4,5,6]. After infecting B-lymphocytes, EBV extends their lifespan, thereby increasing the likelihood of mutations. These mutations can involve alterations in *BCL6* and *MYC* expression, the activation of signaling pathways such as NF-kB, *BCL2*, PI3K/AKT/mTOR, and changes in immunoglobulin switching. These mutations may lead to stepwise advancement from early lesions to polymorphic PTLD and then to clonally expanded monoclonal ones [7,8,9,10,11]. Regarding solid organ transplant recipients, more than 90% of EBV-PTLD cases are attributed to the host, whereas in HCT, the majority of EBV-PTLD cases are of donor origin [12]. PTLD cases ought to be categorized utilizing the tumor classification system for hematopoietic and lymphoid tissues of the World Health Organization 2022 [13]. Hyperplasia, polymorphic or monomorphic post-transplant lymphoproliferative disorders, and classic Hodgkin lymphoma are categorized as “Lymphoid proliferations and lymphomas associated with immune deficiency and dysregulation”. The incidence of PTLD after allogeneic hematopoietic stem cell transplantation is reported to vary between 0.2% and 17% [14,15]. The majority (60%) of cases develop within the first year post-transplantation [14].

Risk factors that have been identified include the transplant type, acute and chronic graft-versus-host disease (GVHD, reduced intensified conditioning (RIC), human leukocyte antigens (HLA)-mismatch, EBV seromismatch (R−/D+), patient age, ex vivo and in vivo T-lymphocyte-depleting treatments (anti-thymocyte globulin (ATG), primary immunodeficiency disorders, splenectomy, and cytomegalovirus reactivation [2,14,16,17,18,19,20,21,22,23,24]. The incidence of PTLD is on the rise, due to the expansion of the types of transplantation like haploidentical HCT [25]. 

PTLD manifests with persistent fever, lethargy, anorexia, signs of bowel perforation, tonsillar enlargement and/or inflammation, symptomatic hepatosplenomegaly, subcutaneous nodules, and gradual decline in peripheral blood cell counts [1,15,26]. Furthermore, patients with PTLD may exhibit headaches or focal neurological deficits [27].

The quantification of viral load is mostly performed via quantitative polymerase chain reaction (Q-PCR); however, there is no agreement on the EBV threshold that should trigger further investigations and pre-emptive therapy [28,29]. This lack of consensus can be attributed to varying sample materials and the absence of standardized PCR techniques [30,31]. In the absence of universally accepted laboratory assays [32,33,34], some researchers working with plasma samples suggest either a threshold of 1000 EBV copies/mL on two consecutive occasions, or 10,000 EBV copies/mL on one sample [22,35]. When whole blood samples are examined, a cutoff of 40,000 EBV copies/mL has been proposed [31,36,37,38], given that EBV load in plasma is 10- to 100-fold lower than in whole blood samples [39]. Guidelines from the European Conference on Infections in Leukemia (ECIL-6) advise that EBV monitoring should be initiated no later than four weeks post-HCT and occur at least once weekly, until cellular immunity has been reconstituted [31].

The types of pre-emptive therapies which can be regularly employed are the reduction of immunosuppressive therapy [40] and the administration of an anti-CD20 monoclonal antibody, rituximab. The use of rituximab resulted in better short-term outcomes in PTLD [41,42,43,44,45,46,47,48,49,50], although concerns exist regarding its long-term side effects. These side effects include B-cell suppression, hypogammaglobulinemia, progressive multifocal leukoencephalopathy, and increased susceptibility to infections [51,52,53,54,55]. The pre-emptive use of rituximab is recommended by American and European guidelines in order to treat rising EBV viral loads in peripheral blood without clinical symptoms, before the diagnosis of EBV-PTLD. However, pre-emptive therapy can lead to the administration of rituximab to patients who would not have developed PTLD. This exposes patients to the potential toxicities associated with the use of this treatment, such as cytopenias and infections. 

Furthermore, noteworthy advancements have been achieved in the field of adoptive immunotherapies for treating PTLD, particularly regarding in vitro autologous donor T-cells, which is practiced as a clinical study in our center, or HLA-matched banked third-party donor polyclonal EBV-specific cytotoxic T-cells (EBVSTs) [56,57,58,59,60,61]. Additionally, chimeric antigen receptor T-cell (CAR-T) therapy has been proven effective in certain cases [62,63,64,65]. Although it has shown some promising outcomes in HCT-PTLD [66], its utilization is limited by the need for immunosuppressive regimens during the initial stages after HCT, which is when HCT-PTLD generally develops. This restriction may impair the effectiveness of CAR-T therapy for treating HCT-PTLD. Moreover, agents like lenalidomide [67,68] and bortezomib [69,70] have demonstrated remarkable outcomes. However, the availability of these treatment methods is often limited [31], despite recent advances [56].

Therefore, it is crucial to investigate approaches to prevent, pre-emptively treat, and eliminate the disease once it manifests. Nonetheless, due to the low incidence of PTLD, there have been limited clinical trials examining such approaches. This article seeks to investigate the occurrence of EBV-PTLD in HCT recipients, focusing on the clinical manifestations and risk factors associated with the disease.

## 2. Results

Out of 546 recipients of hematopoietic cell transplants, clinically relevant EBV reactivation occurred in 100 patients (18%). The majority of these cases had hematologic malignancy (*n* = 98) while 2 had aplastic anemia. Graft sources included matched sibling (*n* = 20), unrelated (*n* = 68), or haploidentical donors (*n* = 12) (Figure 1). Haploidentical donors were significantly higher in patients with clinically relevant EBV reactivation compared to our transplant population (12% versus 6%, *p* < 0.001).

The majority (88/100) of patients received myeloablative conditioning, while 12 received reduced-intensity conditioning. The median time to detection of clinically relevant EBV reactivation was 65 (range 20–2970) days post-transplant, with a median viral load of 24,500 viral genome copies (VGC)/mL (range 8690–2,670,000 VGC/mL) (Appendix A entitled “Patients-Viral loads”). Pre-emptive rituximab was administered to all 74 patients at a median of 4 days post-EBV reactivation, as soon as the reactivation was detected. The majority, 63 out of 74, received one cycle, until the EBV load became undetectable. The number of rituximab cycles (median 1, range 1–3) were not associated with survival outcomes. 

Relapse of clinically relevant EBV reactivation occurred in 13 out of 100 patients, with a higher incidence among those with delayed resolution of infection (27 versus 14 days in non-relapsed patients, *p* < 0.01). Late-onset neutropenia related to rituximab was noted in 16 out of 74 patients and significantly correlated with increased EBV loads. Multivariate analysis confirmed that haploidentical donors *(p* = 0.001), use of ATG *(p* < 0.001) and rituximab-related late-onset neutropenia *(p <* 0.001) were independently associated with increased viral loads. Concurrent cytomegalovirus (CMV) reactivation occurred in 47 patients, with significantly delayed resolution of EBV infections noted in patients receiving pre-emptive anti-CMV treatments, indicating greater immunosuppression. 

Five patients (two with haploidentical and three with unrelated donors) developed PTLD at 41 days post-transplantation. The patients presented fever (5/5), anorexia (5/5), lymphadenopathy (4/5), hepatosplenomegaly (3/5), gradual decline in peripheral blood counts (4/5), and focal neurological deficits (1/5). Receiver operating characteristic (ROC) curve analysis identified a cutoff of 67,150 VGC/mL that predicted PTLD with 80% sensitivity and specificity (see green line in Figure 2).

Relapse-free survival (RFS), overall survival (OS), and treatment-related mortality (TRM) (Appendix A) in the entire cohort were similar regardless of EBV viral load (<50,000 VGC/mL) or PTLD [4-year RFS 32.2%; 4-year OS was 48.1% with a median follow-up of 29 months (4–216)]. Multivariate analysis revealed that ATG and chronic GVHD were independently associated with OS (*p* < 0.001, *p* < 0.05, respectively), while ATG, chronic GVHD and age at transplant were independently associated with higher TRM (HR: 0.1, 1.16, 1.03, 95%CI: 0.15–0.5, 0.008–1.16, 1.007–1.05, respectively, *p* < 0.05). A trend for higher TRM was also noted among patients with EBV loads higher than 50,000 VGC/mL.

## 3. Discussion

In the present study, we investigated the incidence, clinical significance, and relapse rate of EBV reactivation in a large cohort of HCT recipients. With the median time to detection being 65 days post-transplantation, 18% of patients had clinically relevant EBV reactivation. By multivariate analysis, we managed to associate multiple risk factors (late-onset neutropenia, ATG, and haploidentical donors) with increased viral loads. Additionally, CMV coincidental reactivation and subsequent antiviral treatment were correlated with the delayed resolution of EBV infections, most likely due to greater immunosuppression.

In our cohort, PTLD was observed in approximately 1% of patients, a percentage within range of the previously published literature [14,15,17,50]. Notably, this rate was particularly lower than in other studies, where pre-emptive treatment with rituximab was either not a standard of care [71] or utilized a higher viral load threshold for initiating therapy [72]. There was no significant difference in the studied outcomes between responders and non-responders. By ROC curve analysis, we identified a 67,150 viral genomic copies/mL threshold with high sensitivity and specificity for predicting PTLD. Considering the variability observed in the threshold ranges within the literature (ranging from 1000 to 40,000 EBV copies/mL) and depending on the center-specific cutoff values, the application of this threshold has the potential to reduce the unnecessary administration of rituximab and its associated adverse effects in a significant number of patients. In a similar setting, a recent study reported a decreased EBV-DNA clinically significant threshold, but with lower sensitivity and specificity [73].

Considering rituximab’s detrimental effects, we identified late-onset increased neutropenia in 16 patients receiving therapy, leading to increased immunosuppression. Correlation with increased viral loads was observed as expected. Cycles of treatment did not seem to be associated with survival outcomes. Our study aligns with current guidelines [31] that recommend a single infusion, as most patients required only one cycle of rituximab until an undetectable EBV load. Despite concerns about the effects of rituximab on immune reconstitution, a study of 319 consecutive allo-HCTs [52] found that short- and long-term survival were not inferior in patients who received rituximab. This underscores the effectiveness of pre-emptive rituximab, in agreement with our study’s results.

Relapse-free survival, overall survival and treatment-related mortality in the entire cohort were similar regardless of EBV viral load (<50,000 VGC/mL) or PTLD, in agreement with other studies reporting similar results [42,72,74,75]. For instance, Raberahona et al. reported that blood EBV viral load is frequently detectable after HCT but suggests no strong association with survival [42]. Solano et al. found no significant difference in the initial plasma EBV-DNA load among episodes of self-resolving EBV DNAemia, those requiring rituximab treatment or those leading to PTLDs [75]. Moreover, Duver et al. stated that the overall survival of patients with or without viral infections did not differ significantly [74]. However, in our study, ATG and chronic GVHD were independently associated with OS. Finally, ATG, chronic GVHD, age at transplant, and EBV loads above 50,000 VGC/mL were linked with higher TRM, confirming their role as risk factors.

Our study is limited by its single-center retrospective nature. Nevertheless, its strengths include standard operating procedures (SOPs) in a large patient population reflecting international experience. Another limitation, the use of VCG/mL instead of IU/mL as suggested by recent ECIL guidelines for the quantification of EBV-DNA viral loads, has been overcome by the routine use of standard commercial kit for all the patients [31]. 

As demonstrated in our study, regular monitoring and the use of pre-emptive therapy are effective strategies for preventing EBV-related complications, given its rapid and aggressive nature, especially in immunocompromised individuals. A useful cutoff point (67,150 VGC/mL) for the prevention of PTLD has been identified, with 80% specificity and sensitivity. Our study adds valuable data to the relevant literature, helping to address the significant uncertainty around the clinically important EBV levels in these patients. Our findings suggest that our diagnostic and management approach has led to acceptable survival rates in patients with PTLD.

However, our findings can be used by other laboratories as a protocol and not as absolute values. For PLTD, the diagnosis remains clinical. Therefore, our suggestion is that these levels should serve as alerts to clinicians for an extensive investigation.

## 4. Materials and Methods

### 4.1. Study Population

In this retrospective study, we investigated the molecular clinical features of EBV reactivation in patients who underwent hematopoietic cell transplantation at our JACIE (Joint Accreditation Committee-ISCT & EBMT) accredited unit between 2007 and 2019 (Table 1).

### 4.2. Conditioning Regimens—Monitoring

All patients with unrelated or haploidentical donors received rabbit ATG, as previously described [76,77]. To avoid reactions to ATG, methylprednisolone was used on the ATG infusion days. The pre-emptive administration of rituximab was the standard of care in our unit during this period. Treatment with rituximab was received at a scheme determined by the physician according to the dynamics of the infection. The assessment and grading of acute GVHD were performed according to the criteria of Glucksberg et al. [78], while chronic GVHD was assessed and graded according to the 2014 National Health Institute criteria [79,80]. In addition, prophylactic granulocyte-colony stimulating factor (GCSF) was used in all regimens after transplantation. Supportive care also comprised prophylactic platelets transfusion if platelet counts decreased to <20 × 10^9^/L or prophylactic red blood cells transfusion if hemoglobin levels decreased to <8 g/dL. All patients received supportive treatment against bacterial, fungal, and viral infections. Trimethoprim-sulfamethoxazole was used as prophylaxis for *Pneumocystis jirovecii* infection. We measured EBV-DNA every 15 days over a minimum period of 12 months, or longer if immunosuppression was present, in all patients. DNA was extracted from whole blood using a QIAamp^®^ DNA Mini Kit (Qiagen, Hilden, Germany) and EBV- DNA was measured by an artus^®^ EBV RG PCR Kit (Qiagen, Hilden, Germany) on a Rotor-Gene Q MDx Instrument (Qiagen, Hilden, Germany). The analytical detection limit of the assay is 1.02 copies/µL. Clinically relevant EBV reactivation was characterized by the presence of >8500 viral genomic copies (VGC)/mL in whole blood, which was documented during regular molecular monitoring using quantitative real-time PCR. Recurrent infections have been treated with the same protocol. We considered undetectable levels of EBV as indicative of the resolution of infection. Patients with PTLD, confirmed by lymph node biopsy, were managed according to international standards.

### 4.3. Statistical Analysis

Data analysis was carried out using the statistical program SPSS 22.0 (IBM SPSS Statistics for Windows, Version 22.0., Armonk, NY, USA). Descriptive statistics were performed using the median and range for continuous variables and frequency for categorical variables. Continuous variables were assessed for normality and compared using a Mann–Whitney-test or *t*-test. The following factors were studied: age, type of disease/donor/graft, infections, graft-versus-host disease, rituximab cycles, late-onset neutropenia, ATG conditioning, treatment-related mortality, overall, and relapse-free survival. Categorical variables were compared using the chi-square test. The Kaplan–Meier method was used for performing survival analysis, and a log-rank test was utilized for comparison of survival curves. Cox regression analysis was conducted to identify univariate and multivariate predictors of survival, with time-dependent covariates computed through SPSS analysis. Statistical significance was assessed by the Gray test and Fine and Gray regression modeling. The level of statistical significance was defined at 0.05. 

## 5. Conclusions

Our study investigated the incidence, clinical significance and risk factors associated with EBV reactivation in a substantial cohort of HCT recipients. 18% of the patients experienced clinically relevant EBV reactivation around 65 days post-transplantation. Multivariate analysis allowed us to identify crucial risk factors such as ATG administration, late-onset neutropenia, and haploidentical donor grafts, which all correlated with elevated viral loads. The observed PTLD incidence of (1%) aligns with the previous literature. Furthermore, we proposed a practical threshold of 67,150 VGC/mL with high sensitivity and specificity for PTLD prediction in an attempt to address the uncertainty regarding clinically significant EBV levels in these patients. Despite the inherent limitations, our findings underscore the significance of regular monitoring and pre-emptive therapy for managing EBV-related complications, particularly within the context of immunocompromised individuals. 

## Figures and Tables

**Figure 1 ijms-24-16029-f001:**
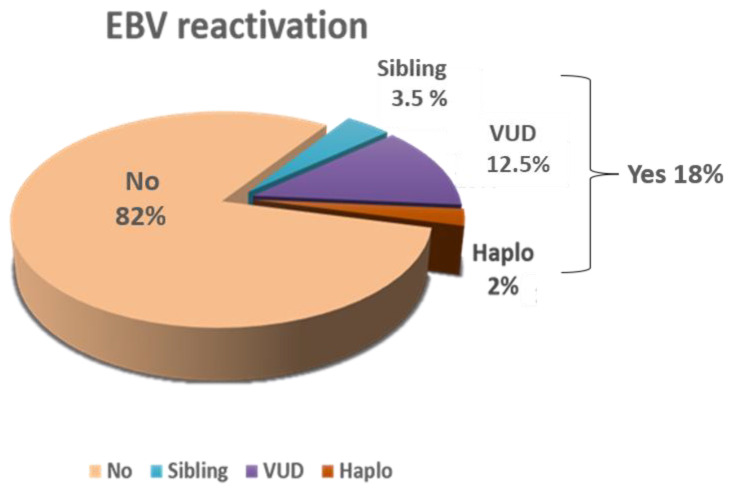
Patient sampling scheme employed in the study.

**Figure 2 ijms-24-16029-f002:**
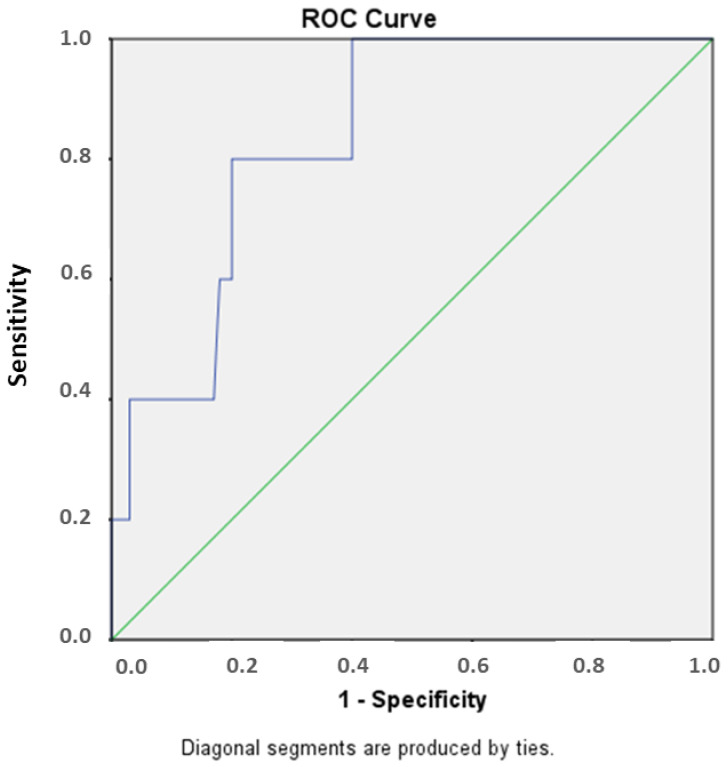
A cutoff of 67,150 copies predicted post-transplant lymphoproliferative disease (PTLD) with 80% sensitivity and specificity.

**Table 1 ijms-24-16029-t001:** Descriptive table of 100 patients with clinically relevant Epstein–Barr virus (EBV) reactivation. Mean and median viral genome copies (VGC)/mL were calculated from all positive pre-treatment samples.

Biographical Data	Category	Frequency	Percentage	Mean (Copies/mL)	Median (Copies/mL)	Min (Copies/mL)	Max (Copies/mL)
Gender	Male	56/100	56%	112,944	27,950	9120	2,760,000
	Female	44/100	44%	102,348	20,000	8690	1,150,000
Age group	0–18	6/100	6%	64,567	17,700	10,400	180,000
	18–44	49/100	49%	118,480	27,200	8700	2,760,000
	44+	45/100	45%	103,005	21,000	8690	1,150,000
Type of transplant	Sibling	20/100	20%	95,622	20,500	9930	1,150,000
	VUD	68/100	68%	74,100	22,850	8690	951,000
	Haplo	12/100	12%	323,075	48,900	10,800	2,760,000
Type of conditioning	Myeloablative	88/100	88%	118,796	27,250	8690	2,760,000
	RIC	12/100	12%	19,225	17,250	8700	47,100
aGVHD	Yes	64/100	64%	95,706	26,650	8690	2,760,000
	No	36/100	36%	130,637	22,750	9930	1,150,000
cGVHD	Yes	82/100	82%	76,344	26,650	8690	1,150,000
	No	18/100	18%	253,772	17,550	10,000	2,760,000
Rituximab treatment	Yes	74/100	74%	135,788	30,400	8690	2,760,000
	No	26/100	26%	29,996	16,350	10,300	148,000
Rituximab cycles	One	64/74	86.5%	138,663	29,300	8690	2,760,000
	Two	9/74	12.1%	83,244	29,800	10,400	499,000
	Three	1/74	1.4%	383,000	383,000	383,000	383,000
PTLD	Yes	5/100	5%	684,680	78,400	31,000	2,760,000
	No	95/100	95%	77,495	21,000	8690	1,150,000
EBV relapse	Yes	13/100	13%	281,323	40,000	11,000	2,760,000
	No	87/100	87%	76,339	20,500	8690	1,150,000
CMV reactivation	Yes	47/100	47%	82,766	20,300	8690	1,150,000
	No	53/100	53%	130,778	24,600	9120	2,760,000

## Data Availability

Data will be immediately available upon reasonable request.

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
