# Peer review of "Pre-Emptive Use of Rituximab in Epstein–Barr Virus Reactivation: Incidence, Predictive Factors, Monitoring, and Outcomes"

_ijms, 2023, doi:10.3390/ijms242216029_

Round 1
Reviewer 1 Report (New Reviewer)
Comments and Suggestions for Authors
The study revealed clinical traits of PTLD of hematopoietic cell transplantation caused by the EBV. Its communication form suits the necessity in timely presentation to the research field, however, there are limitations and many concerns should be carefully addressed.
#1 the rationale in study design needs amending. As authors say "Relapse-free survival, overall survival and treatment-related mortality among the entire cohort were similar regardless of the EBV viral load or PTLD, in agreement with other studies reporting similar results", what is the point of this research would be questioned. It is not sufficient for novelty just for curiosity in a bunch of data collection.
2# the methods used in the study led to concerns in evidence reliability, especially in viral load detection. If authors want to claim any preferable cut-off value in comparison with exist one, they need to present result based on identical method or at least an acceptable normalization standard. Otherwise, the new observations would lead to nowhere and refer to nothing.
3# From data presented in paper, it almost circles around 100 patients of EBV reaction. It is true that the whole cohort consists of more than 500 subjects, but the strengths from only 100 subjects would be very limited.
4# Moreover, the single-center retrospective nature really damages the unbiased principle. According to current evidence, it could only arouse some interest in the field instead of suggesting anything. Authors needs more convincible data or comparable observation to support their conclusions.
Small tips on language,
In the 2nd para. of Discussion section, referee thought that the coma "," following (ranging from 1000 to 40,000 EBV copies/mL) should be omitted and replaced by "and".
Comments on the Quality of English LanguageIn general, the language use is acceptable. Small problems remain in punctuation.
Author Response
Please see the attachment.

Reviewer 2 Report (New Reviewer)
Comments and Suggestions for Authors
Paper by Papalexandri A et al. presents data from monitoring of EBV reactivation, EBV- related post-transplant lymphoproliferative disease (PTLD) occurrence and prevention in a group of 546 HSCT recipients. They document that pre-emptive rituximab therapy of high viral load- associated EBV reactivations was effective, resulting in low incidence of PTLD in their cohort. Considering large sample size, appropriate methods used inclusive of adequate statistical analysis the results obtained are valuable and worth publishing. The paper is well written, includes extensive Introduction and Discussion. However some issues should be better explained or amended. Also, several statements need rewording.
Comments:
Material and methods:
Measures for decision for pre-emptive rituximab therapy should be better specified. Why 25% of the patients with clinically relevant EBV reactivation ( i.e. with loads exceeding 8500 VGC/mL) were not treated? Which other issues ( i.e. dynamics of the infection, other defined risk factors) were considered? Were the recurrent infections repeatedly treated?
Use „EBV reactivation“ only for the infections with high viral load in blood is not correct: EBV infection with low viral load is also reactivation. Clinically relevant reactivation should be more appropriate in this context.
How a Mean or Median VGC/mL in Table 1 was calculated? A mean from the all positive samples or from a peak of load? Were only pre-treatment or all the samples calculated?
Which GVHD prophylaxis was used in the cohort studied? Was ATG administered to all the patients enrolled?
Results:
Which outcomes were mentioned in association with the number of rituximab cycles? ( see also the Discussion)
The data on multivariate analysis of viral load and CMV coinfection are not shown.
Patients with PTLD did not receive pre-emptive therapy or were refractory to it? From which samples the cut-off predictive value 67150 VCG/mL was calculated? Did they precede PTLD development or not?
Discussion:
OS, RFS and TRM in the cohort studied are discussed but the data are not shown.
In the Discussion, some statements are contradictory: see par. 3: „..... EBV loads above 50 000 linked with TRM“, par.4 .“...increased viral loads or PTLD were not associated with poor survival“.
Author Response
Please see the attachment

Reviewer 3 Report (New Reviewer)
Comments and Suggestions for Authors
The article concerns incidence of posttransplant lymphoproliferative disease (PTLD), multifactorial assessment of potential therapeutic risk, and optimal monitoring regimens for Epstein-Barr virus (EBV) after hematopoietic stem cell transplants (HCT), performed, mostly, in oncohematological diseases with myeloablative conditioning. The statistical evidence is based on a representative group of HCTs (n=546), with 5 proven PTLD cases. The multivariate analysis of multiple risk factors post-HCT was performed in conventional mode. The authors have evaluated common clinical effects and virological consequences of pre-emptive treatment with rituximab, a potent suppressor of B-cell immunity. The latter finding seems to be the most interesting result of study.
Remarks:
Abstract: The definition of “PTLD caused by the Epstein-Barr virus (EBV)” could be (more carefully) replaced by “PTLD associated with EBV”.
The cutoff level of 67,150 viral copies/mL for predicting PTLD seems to be rather crude. This level is based only on 5 cases of this rare complication. Hence, this assumed value should be later specified in more extended studies.
Results (lane 1, 5, 7 and further on): The term “clinically relevant EBV reactivation” needs better definition (either by own clinical criteria, or by literature sources referred).
Page 3 (line 1, just under the diagram. The terms of PTLD diagnosis in these 5 cases should be also provided, since PTLD may develop at quite late terms after HSCT.
A clear conclusion should be made (Page 7, bottom lanes).
Page 5, paragraph 2: Rituximab-associated neutropenia is mentioned in Discussion. Therefore, extent and ranges of this side effect (as well as potential lymphopenia) should be also mentioned under Results.
Page 8 (Materials and methods): Lane 2-3: one should clarify (similar to Table 1) the real number of patients subjected to pre-emptive rituximab treatment, as well as median number of courses and median terms of RTX therapy (months/weeks post-transplant).
Some misprints are found. For example, page 2, Para 1, last line …within first year of transplantation… should be replaced by …within first year post-transplant… Page 5, para 4, lane 2: …real-world evidence… may be replaced by …international experience…
Comments on the Quality of English LanguageMinor copy editing is required
Round 2
Reviewer 1 Report (New Reviewer)
Comments and Suggestions for Authors
Authors failed to address referee's concerns and did not show willingness to face the issue pointed out in last session. Especially, the data comparison availability and the limitation of the study design which were critical to this study remains serious flaws. It cannot be acceptable until authors get serious to those issues.
Comments on the Quality of English LanguageNA
Author Response
We would like to thank the Reviewer for the comments. We have added the statistical analysis about OS, RFS and TRM as supplemental files. As per our study, we have to insist that the value of real-world evidence from large cohorts, such as ours, has been worldwide recognized, especially since there are very limited prospective multicenter studies in the whole transplant field.
Reviewer 2 Report (New Reviewer)
Comments and Suggestions for Authors
The comments were acctepted or sufficiently cleared. I recommend to add missing analyses (comment 6 and 8) as supplemental files.
Author Response
We would like to thank again the reviewer for the fruitful comments that helped us improve our manuscript. We have added as supplemental files the analysis about OS, RFS and TRM. In addition, we have added in the manuscript the p-values from the multivariate analysis of haploidentical donors (p=0.001), anti-thymocyte globulin (ATG) use (p<0.001), and late-onset neutropenia related to rituximab (p<0.001) associated with increased viral load.
This manuscript is a resubmission of an earlier submission. The following is a list of the peer review reports and author responses from that submission.
Round 1
Reviewer 1 Report
Comments and Suggestions for Authors
Dear authors,
I enjoyed reading a paper entitled: Pre-emptive Use of Rituximab in Epstein-Barr Virus Reactivation: Incidence, Predictive Factors, Monitoring, and Outcomes, but I have several suggestions for improving the overall quality of it.
1. Lines 167, 169, and 171 references need to be corrected. The author's names are only mentioned. There is no number of reference
2. Material and methods
- You need to mention which sample was used for EBV quantitation. Was it just one sample per patient or more? How often were these samples taken?
What protocol and kit were used for EBV quantification, and was always the same protocol and kit used for all patients? What is the sensitivity of using RTQ-PCR for EBV detection?
Term RQ-PCR replaces with RTQ-PCR or quantitative real-time PCR.
I think patients' names should be omitted and replaced with some ID numbers in the table regarding non-published material.
Overall nicely written paper with some additional information regarding EBV and PTLD.
All the best
Reviewer 2 Report
Comments and Suggestions for Authors
The authors conducted a retrospective study on the occurrence of EBV-associated post-transplant lymphoproliferative disease (PTLD) among 546 Hematopoietic Cell Transplantation (HCT) recipients. The study focused on the disease's risk factors and clinical symptoms. The study also emphasizes the importance of frequent, close monitoring of post-transplant complications for early intervention to enhance survival rates and reduce mortality.
Among the 546-study cohort, only 1% of patients presented with PTLD. The authors found that 67150 viral genomic copies/ml cut-off represents a storage prediction for PTLD. The study finding suggests that PTLD, anti-thymocyte globulin (ATG), and graft-versus-host disease (GVHD) were independently associated with unfavorable outcomes.
I only have minor comments.
Minor comments:
- Spell out GVHD when the acronym is first mentioned in line 14
- Spell out ATG when the acronym is first mentioned in line 22
- I suggest that the authors be consistent in spelling the word pre-emptive with/out the dash (pre-emptive therapy or preemptive therapy)
- Spell out ROC (Receiver Operating Characteristic) in line 126
Reviewer 3 Report
Comments and Suggestions for Authors
In this manuscript, Authors described investigated the occurrence of EBV-PTLD in 546 HCT recipients, focusing on clinical manifestations and risk factors associated with the disease and managed to identify 67150 viral genomic copies/ml as the cut-off point for predicting PTLD. According to this study, prophylactic measures such as regular monitoring, preemptive therapy, and supportive treatment against infections can be proven effective in preventing EBV-related complications. This is a well-written manuscript, the author made a great effort, but I suggest some changes that must be made to the text, to contribute to a better understanding of the points they are trying to make.
Major comments
1. In line 106, the author investigated the occurrence of EBV-PTLD in 546 HCT recipients. EBV reactivation occurred in 100 patients (18%). Rituximab was given to 74 out of 100 EBV positive recipients. The author can make a chart to represent sampling, so that it will be clearer for the readers.
2. In line 148, Author said 1% of patients presented PTLD. It is a very simple study; the author must make some more comparison between recovered or non-recovered EBV reactivation recipients.
3. In there any correlation between single dose and multiple doses of rituximab with EBV-PTLD.
4. The author did not site this paper, this paper is like this study. “Pre-emptive rituximab treatment for Epstein–Barr virus reactivation after allogeneic hematopoietic stem cell transplantation is a worthwhile strategy in high-risk recipients: a comparative study for immune recovery and clinical outcomes.”
Comments on the Quality of English LanguageMinor editing of English language is required.
Round 2
Reviewer 3 Report
Comments and Suggestions for Authors
Overall, the manuscript is improved and I am accepting paper in present form.
Author Response
We would like to thank the Reviewer for the valuable comments that have helped us significantly improve our Communication.